# MeshAnything: Artist-Created Mesh Generation with Autoregressive Transformers

**Yiwen Chen**[1,2][*], **Tong He**[2][†], **Di Huang**[2], **Weicai Ye**[2], **Sijin Chen**[3], **Jiaxiang Tang**[4]
**Xin Chen**[5], **Zhongang Cai**[6], **Lei Yang**[6], **Gang Yu**[7], **Guosheng Lin**[1][†], **Chi Zhang**[8][†]
[1]S-Lab, Nanyang Technological University    [2]Shanghai AI Lab
[3]Fudan University    [4]Peking University    [5]University of Chinese Academy of Sciences
[6]SenseTime Research    [7]Stepfun    [8]Westlake University
**https://buaacyw.github.io/mesh-anything/**

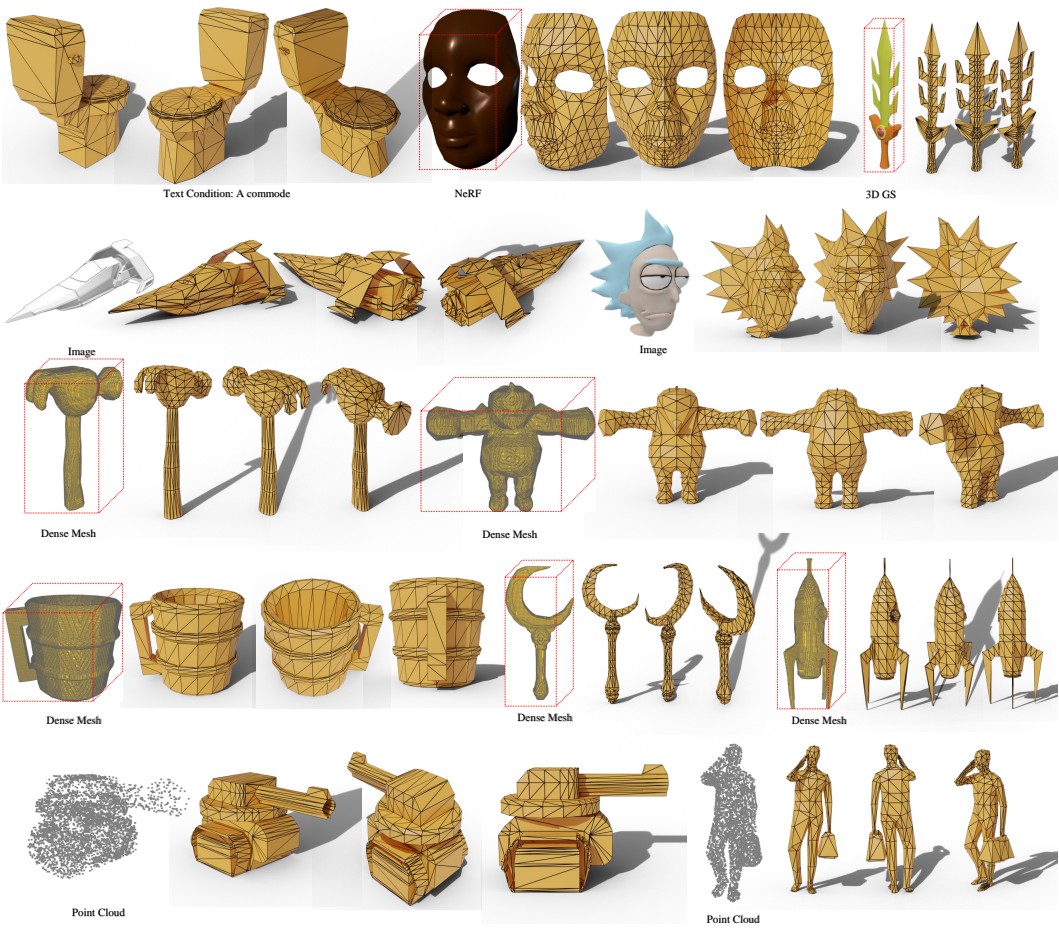

Figure 1: **MeshAnything converts different 3D representation into Artist-Created Meshes (AMs), i.e., meshes created by human artists.** It can be combined with various 3D asset production pipelines, such as 3D reconstruction and generation, to transform their results into AMs that can be seamlessly applied in the 3D industry.

## Abstract

Recently, 3D assets created via reconstruction and generation have matched the quality of manually crafted assets, highlighting their potential for replacement. However, this potential is largely unrealized because these assets always need to

---

[*]Work done during a research internship at Shanghai AI Lab.
[†]Corresponding Authors.

be converted to meshes for 3D industry applications, and the meshes produced by current mesh extraction methods are significantly inferior to Artist-Created Meshes (AMs), i.e., meshes created by human artists. Specifically, current mesh extraction methods rely on dense faces and ignore geometric features, leading to inefficiencies, complicated post-processing, and lower representation quality. To address these issues, we introduce MeshAnything, a model that treats mesh extraction as a generation problem, producing AMs aligned with specified shapes. By converting 3D assets in any 3D representation into AMs, MeshAnything can be integrated with various 3D asset production methods, thereby enhancing their application across the 3D industry. The architecture of MeshAnything comprises a VQ-VAE and a shape-conditioned decoder-only transformer. We first learn a mesh vocabulary using the VQ-VAE, then train the shape-conditioned decoder-only transformer on this vocabulary for shape-conditioned autoregressive mesh generation. Our extensive experiments show that our method generates AMs with hundreds of times fewer faces, significantly improving storage, rendering, and simulation efficiencies, while achieving precision comparable to previous methods.

# 1 INTRODUCTION

In recent years, the 3D community has experienced rapid advancements, with a variety of methods developed for automatically producing high-quality 3D assets. These methods, including 3D reconstruction (Mildenhall et al., 2020; Yu et al., 2021; Barron et al., 2021; 2022; Kerbl et al., 2023b; Huang et al., 2024), 3D generation (Poole et al., 2023; Liu et al., 2023a; Wang et al., 2023; Long et al., 2023; Sun et al., 2023; Hong et al., 2023; Tang et al., 2024; Xu et al., 2024; Wei et al., 2024), and scanning (Daneshmand et al., 2018; Haleem & Javaid, 2019; Haleem et al., 2022), can produce 3D assets with shape and color quality comparable to manually created ones. The success of these methods reveals the potential to replace manually created 3D models with automatically produced ones in the 3D industry, including applications in games, movies, and the metaverse, significantly reducing time and labor costs.

However, this potential remains largely unrealized because the current 3D industry predominantly relies on mesh-based pipelines for their superior efficiency and controllability, while methods for producing 3D assets typically use alternative 3D representations to achieve optimal results across various scenarios. Therefore, substantial efforts (Lorensen & Cline, 1987; Chernyaev, 1995; Lorensen & Cline, 1998; Shen et al., 2021b; Chen et al., 2022; Shen et al., 2023) are devoted to converting other 3D representations into meshes and have achieved some success. Meshes produced by these methods approximate the shape quality of those created by human artists, which we refer to as Artist-Created Meshes (AMs), but they still fall short in addressing the aforementioned issues.

This is because all meshes produced by these methods (Lorensen & Cline, 1987; Chernyaev, 1995; Lorensen & Cline, 1998; Shen et al., 2021b; Chen et al., 2022; Shen et al., 2023) exhibit significantly poorer topology quality compared to AMs. As shown in Fig. 2, these methods rely on dense faces to reconstruct 3D shapes, completely ignoring geometric characteristics. Using these meshes in the 3D industry leads to three significant problems: First, converted meshes typically contain several orders of magnitude more faces compared to AMs, leading to significant inefficiencies in storage, rendering, and simulation. Moreover, the converted meshes complicate post-processing and downstream tasks in the 3D pipeline. They significantly increase the challenge for human artists in optimizing these meshes due to their chaotic and inefficient topologies. Finally, previous methods struggle to represent sharp edges and flat surfaces, resulting in oversmoothing and bumpy artifacts as shown in Fig. 2.

In this work, we aim to solve the aforementioned issues to facilitate the application of automatically generated 3D assets in the 3D industry. As mentioned earlier, all previous methods (Lorensen & Cline, 1987; Chernyaev, 1995; Lorensen & Cline, 1998; Shen et al., 2021b; Chen et al., 2022; Shen et al., 2023) extract 3D meshes with excessively dense faces in a reconstruction manner, which inherently cannot solve these issues. Therefore, we diverge from previous approaches by formulating mesh extraction as a generation problem for the first time: we teach models to generate Artist-Created Meshes (AMs) that are aligned with the given 3D assets. The meshes generated by our

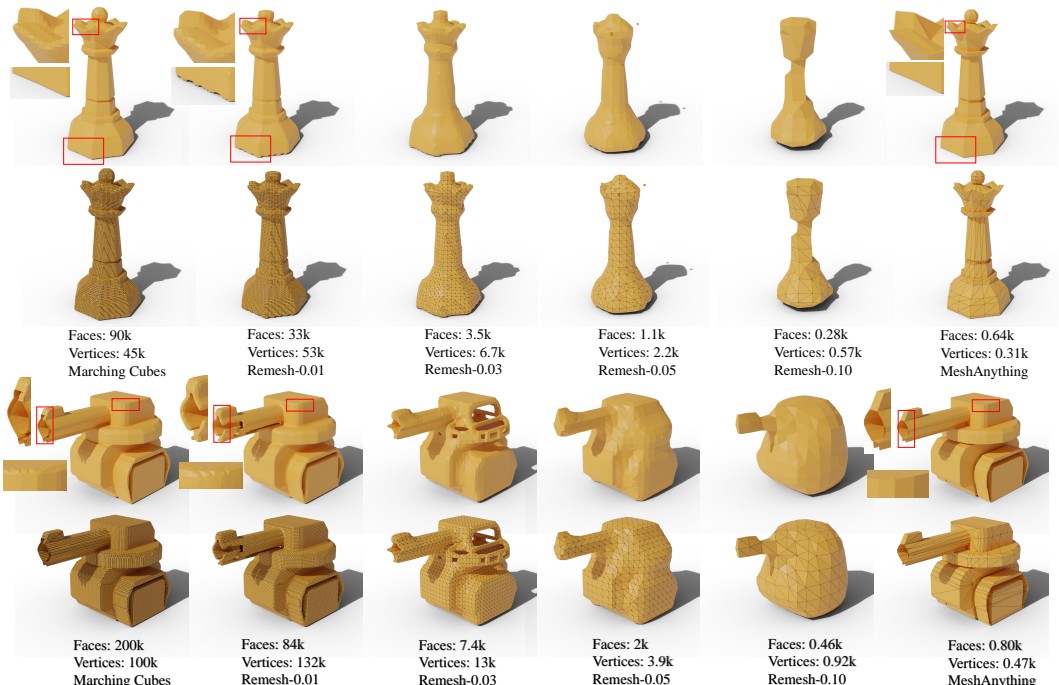

Figure 2: **Comparison with Marching Cubes Lorensen & Cline (1987) and Remesh Blender Development Team (2024).** We apply Marching Cubes and MeshAnything to ground truth shapes and then apply remeshing to the Marching Cubes results with different voxel sizes. Existing methods extract meshes in a reconstruction manner, ignoring the geometric features of the object and producing dense meshes with poor topology. These methods fundamentally fail to capture sharp edges and flat surfaces, as shown in the zoomed-in figure.

methods mimic the shape and topology quality of those created by human artists. Consequently, our setting, namely Shape-Conditioned AM Generation, is fundamentally free from all previous issues, enabling seamless integration of the generated results into the 3D industry pipeline.

However, training such a model presents significant challenges. The first challenge is constructing the dataset, as we need paired shape conditions and Artist-Created Meshes (AMs) for model training. The shape condition must be efficiently derived from as many diverse 3D representations as possible to serve as a condition during inference. Additionally, it must have sufficient precision to accurately represent 3D shapes and be efficiently processed into features that can be injected into the model. After weighing the trade-offs, we chose point clouds due to their explicit and continuous representation, ease of derivation from most 3D representations, and the availability of mature point cloud encoders (Qi et al., 2017a;b; Zhao et al., 2024).

We filter out high-quality AMs from Objaverse (Deitke et al., 2023b;a) and ShapeNet (Chang et al., 2015). When obtaining paired shape conditions, a naive approach would be to sample point clouds directly from AMs. However, this leads to poor results during inference because the sampled point clouds have excessive precision, while automatically produced 3D assets cannot provide point clouds of similar quality, causing a domain gap between training and inference. To address this issue, we intentionally corrupt the shape quality of AMs. We first extract the signed distance function from AMs (Wang et al., 2022), convert it into a coarser mesh using (Lorensen & Cline, 1987), and then sample point clouds from this coarse mesh to narrow the domain gap in shape conditions between inference and training.

Following (Siddiqui et al., 2023), we use a VQ-VAE (Van Den Oord et al., 2017) to learn a mesh vocabulary and train a decoder-only transformer (Vaswani et al., 2017) on this vocabulary for mesh generation. To inject shape condition, we draw inspiration from the recent success of multimodal large language models (MLLM) (Wu et al., 2023; Liu et al., 2024b), where image features encoded by pre-trained image encoders are projected into the token space of the large language models for efficient multimodal understanding. Similarly, we treat the mesh tokens obtained from the trained

VQ-VAE as the language token in LLMs and use a pre-trained encoder (Zhao et al., 2024) to encode the point clouds into shape features, which is later projected into the mesh token space. These shape tokens are placed at the beginning of the mesh token sequences, effectively serving as the shape conditions for next-token predictions. After predictions, these predicted mesh tokens are decoded back to meshes with the VQ-VAE decoder (Siddiqui et al., 2023).

To further enhance the quality of mesh generation, we develop a novel noise-resistant decoder for robust mesh decoding. Our observation is that as the decoder in the VQ-VAE (Van Den Oord et al., 2017) is only trained with ground truth token sequences from the encoder, it could potentially lead to a domain gap when decoding the generated token sequences. To mitigate this problem, we inject the shape condition into the VQ-VAE decoder as auxiliary information for robust decoding and fine-tune it after the VQ-VAE training. This fine-tuning process involves adding noise to the mesh token sequences to simulate possible poor-quality token sequences from the decoder-only transformer, thus making the decoder robust to such poor-quality sequences.

Finally, we introduce our model, MeshAnything, trained based on the aforementioned techniques. As shown in Fig. 1, MeshAnything can convert 3D assets across various 3D representations into AMs, thereby significantly facilitating their application. Furthermore, our extensive experiments demonstrate that our method generates AMs with significantly fewer faces and more refined topology, while achieving precision metrics that are close to or comparable with previous methods.

In summary, our contributions are as follows:

- We highlight one important reason why current automatically produced 3D assets cannot replace those created by human artists: current methods cannot convert these 3D assets into Artist-Created Meshes (AMs). To solve this issue, we propose a novel solution called Shape-Conditioned AM Generation, which aims to generate AMs aligned with given shapes.
- We introduce MeshAnything for Shape-Conditioned AM Generation. MeshAnything can be integrated with various 3D asset production methods, converting their results into AMs to facilitate their application in the 3D industry.
- We develop a novel noise-resistant decoder to enhance mesh generation quality. We inject the shape condition into the decoder as auxiliary information for robust decoding and fine-tune it using noised token sequences to narrow the domain gap between training and inference.
- Extensive experiments demonstrate that Shape-Conditioned Mesh Generation is a more suitable setting for mesh generation, and MeshAnything significantly surpasses previous mesh generation methods.

## 2 RELATED WORKS

### 2.1 MESH EXTRACTION

Methods for extracting meshes from 3D models are numerous and have been a subject of research for decades. Following (Shen et al., 2023), we categorize these methods into two main types: Isosurface Extraction (Lorensen & Cline, 1987; Bloomenthal, 1988; Chernyaev, 1995; Bloomenthal & Bajaj, 1997; Lorensen & Cline, 1998; Chen et al., 2022) and Gradient-Based Mesh Optimization (Chen et al., 2019; Gao et al., 2020; Hanocka et al., 2020; Kato et al., 2018; Shen et al., 2021a; Liao et al., 2018; Shen et al., 2023).

Traditional isosurface extraction methods (Lorensen & Cline, 1987; 1998; Chernyaev, 1995; Doi & Koide, 1991; Ju et al., 2002; Schaefer et al., 2007; Chen & Zhang, 2021; Chen et al., 2022) focus on extracting a polygonal mesh that represents the level set of a scalar function, an area that has seen extensive study in various fields. The most popular method among them is Marching Cubes (Lorensen & Cline, 1987). It divides the space into cells, within which polygons are created to approximate the surface.

Transitioning to more recent developments, the advent of machine learning has ushered in new techniques for generating 3D meshes (Chen et al., 2019; Gao et al., 2020; Hanocka et al., 2020; Kato et al., 2018; Shen et al., 2021a; Liao et al., 2018; Shen et al., 2023). This line of work explores

using neural networks to generate 3D meshes, where the network parameters are optimized through gradient-based methods under specific loss functions.

However, these approaches fundamentally differ from ours. They ignore the characteristics of the shape and inherently cannot produce meshes with efficient topology. In contrast, MeshAnything formulates mesh extraction as a generation problem for the first time, aiming to mimic human artists in mesh extraction and thereby generating Artist-Created Meshes (AMs) with hundreds of times fewer faces.

## 2.2 3D MESH GENERATIONS

3D mesh generation can be mainly divided into two categories: generating dense meshes similar to those produced by previous mesh extraction methods, and generating Artist-Created Meshes (AMs).

The former category is currently the mainstream research focus. Methods such as (Gao et al., 2022; Wei et al., 2024; Xu et al., 2024) directly generate meshes in a feed-forward manner, but because they produce dense meshes with low-quality topology similar to previous mesh extraction methods, they still encounter the same issues when applied in the 3D industry.

Notably, numerous 3D generation methods (Poole et al., 2023; Tang et al., 2023b; Wang et al., 2023; Chen et al., 2024b; Tang et al., 2023a; Yang et al., 2023; Hong et al., 2023; Fang et al., 2023; Chen et al., 2023a; Liu et al., 2024c; Shi et al., 2023; Li et al., 2023; Chen et al., 2023b; 2024c; Tang et al., 2024; Wang et al., 2024; Tochilkin et al., 2024; Liu et al., 2024a; 2023c; Zhang et al., 2024) can also produce meshes. These methods first generate 3D assets and then convert them to dense meshes using mesh extraction methods like (Lorensen & Cline, 1987). Consequently, they face challenges when applied to the 3D industry due to their inefficient topology.

Recently, several works have focused on the second category: generating Artist-Created Meshes(AMs) (Nash et al., 2020; Alliegro et al., 2023; Siddiqui et al., 2023; Chen et al., 2024a). Although our approach also focuses on AM generation, it fundamentally differs from these methods. Since they lack shape conditioning, these methods must simultaneously learn the complex 3D shape distribution—which typically alone requires extensive training (Hong et al., 2023; Tang et al., 2024)—and the topology distribution of AMs, leading to very challenging training processes. In contrast, our methods eliminate the challenge of learning the shape distribution, allowing the model to focus on learning the topology distribution. This not only significantly reduces training costs but also enhances the model's application value.

Among these methods, the most relevant to ours is MeshGPT (Siddiqui et al., 2023), as we follow its architecture. (Siddiqui et al., 2023) introduced a combination of a VQ-VAE (Van Den Oord et al., 2017) and an autoregressive transformer architecture. It first learns a mesh vocabulary with the VQ-VAE and then trains the transformer on the learned vocabulary for mesh generation. However, MeshGPT's results are limited to several categories in ShapeNet. MeshGPT requires a training GPU hours similar to ours, but our method can generalize to unlimited categories in Objaverse. As shown in Fig. 3, this is largely due to the difference in target complexity caused by MeshGPT needing to additionally learn the complex 3D shape distribution.

## 3 SHAPE-CONDITIONED AM GENERATION

In this section, we first introduce the formal formulation for Shape-Conditioned AM Generation and compare it with previous mesh generation settings (Nash et al., 2020; Siddiqui et al., 2023; Alliegro et al., 2023). We show that it can achieve better performance and a broader range of applications compared to the settings in previous mesh generation methods, with significantly less training effort.

Shape-Conditioned AM Generation targets to estimate a conditional distribution $p(\mathcal{M}|\mathcal{S})$. In this formula, $\mathcal{M}$ refers to the Artist-Created Mesh (AM), i.e., the mesh manually modeled by human artists. $\mathcal{S}$ refers to the 3D shape information that indicates the 3D shape to which $\mathcal{M}$ should align. The input form of $\mathcal{S}$ can be diverse, such as voxels or point clouds. Therefore, this versatility allows our method to be integrated with any 3D pipeline that outputs $\mathcal{S}$, such as 3D reconstruction (Mildenhall et al., 2020; Kerbl et al., 2023b), generation (Poole et al., 2023; Hong et al., 2023), and scanning, making these methods more efficient for the 3D industry.

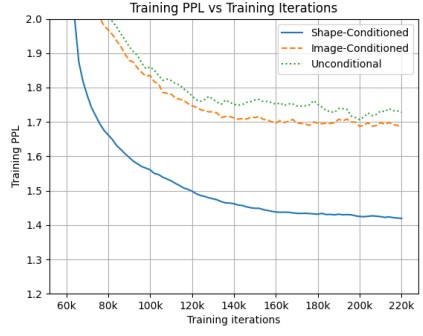
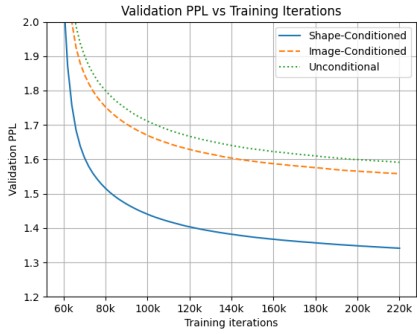

(a) Training Perplexity (PPL)   (b) Validation Perplexity (PPL)

Figure 3: **Training and validation perplexity (PPL) for the mesh generation model under different input conditions.** All models are trained with the same settings as detailed in Section 5.2. The training and validation PPL of shape-conditioned mesh generation is significantly lower than that of unconditional and image-conditioned mesh generation. This indicates that the training burden of shape-conditioned mesh generation is much lower since it avoids learning the complex 3D shape distribution.

Compared to existing AM generation work, they directly estimate the distribution $p(\mathcal{M}|\mathcal{C})$, where $\mathcal{C}$ denotes conditions such as images, text or empty sets for unconditional generation. However, estimating $p(\mathcal{M}|\mathcal{C})$ requires an understanding of both the underlying shape, i.e., $\mathcal{S}$, and complex topological structures $\mathcal{M}$. Given this, we made the following approximation:

$$p(\mathcal{M}|\mathcal{C}) \approx p(\mathcal{M}, \mathcal{S}|\mathcal{C}). \tag{1}$$

According to the chain rule, we have:

$$p(\mathcal{M}, \mathcal{S}|\mathcal{C}) = p(\mathcal{M}|\mathcal{S}, \mathcal{C}) \cdot p(\mathcal{S}|\mathcal{C}). \tag{2}$$

For distribution $p(\mathcal{M}|\mathcal{S}, \mathcal{C})$, given that $\mathcal{S}$ is a much stronger and more direct condition than $\mathcal{C}$, we can make the following approximation:

$$p(\mathcal{M}|\mathcal{S}, \mathcal{C}) \approx p(\mathcal{M}|\mathcal{S}). \tag{3}$$

Combining 1, 2 and 3:

$$p(\mathcal{M}|\mathcal{C}) \approx p(\mathcal{M}|\mathcal{S}) \cdot p(\mathcal{S}|\mathcal{C}), \tag{4}$$

in which $p(\mathcal{M}|\mathcal{S})$ is the focus of our shape-conditioned mesh generation. As shown in Fig. 3, estimating $p(\mathcal{M}|\mathcal{S})$ is much more simpler than $p(\mathcal{M}|\mathcal{C})$, proving that our setting is much easier to train than settings in privous methods.

As for $p(\mathcal{S}|\mathcal{C})$, In the 3D community, numerous large models (Team, 2024; Tang et al., 2024; Xu et al., 2024; Siddiqui et al., 2023) aim to estimate using various 3D representations and demonstrate excellent results. Besides, some single scene 3D asset production methods (Mildenhall et al., 2020; Kerbl et al., 2023b; Barron et al., 2021; 2022; Poole et al., 2023; Liu et al., 2023b; Sun et al., 2023) can also provide samples from this distribution. By integrating our framework with these existing methods, we can leverage their capabilities to enhance our mesh generation process. This integration allows for a more resource-efficient way to estimate $p(\mathcal{M}|\mathcal{C})$, significantly reducing the complexity and resources required compared to previous methods.

## 4 METHOD

In this section, we detail our shape condition strategy in Section 4.1. After that, we provide a detailed description for MeshAnything, which consists of a VQVAE with our newly proposed noise-resistant decoder (Section 4.2) and a shape-conditioned autoregressive transformer (Section 4.3).

Figure 4: **Pipeline Overview.** We introduce MeshAnything, an autoregressive transformer capable of generating Artist-Created Meshes that adhere to given 3D shapes. During training, we inject point clouds features into a decoder-only transformer and supervise it using token sequences derived from the Artist-Created meshes. After training, MeshAnything takes point clouds sampled from various 3D representations as input and generates aligned Artist-Created meshes.

## 4.1 SHAPE ENCODING FOR CONDITIONAL GENERATION

We begin by describing our shape condition strategy. MeshAnything targets learning $p(\mathcal{M}|\mathcal{S})$, so we need to pair each mesh $\mathcal{M}$ with a corresponding $\mathcal{S}$, i.e., the shape condition. Choosing an appropriate 3D representation for $\mathcal{S}$ is non-trivial and should satisfy the following conditions:

1. It should be easily extracted from various 3D representations. This ensures that the trained models can be integrated with a wide range of 3D asset production pipelines (Mildenhall et al., 2020; Kerbl et al., 2023b; Hong et al., 2023; Poole et al., 2023; Tang et al., 2024).

2. It should be suitable for data augmentation to prevent overfitting. To ensure the effectiveness of $\mathcal{S}$ during training, any data augmentation applied to $\mathcal{M}$ must be equivalently applicable to $\mathcal{S}$.

3. It should be efficiently and conveniently input into the model as a condition. To ensure the model comprehends the shape information and to maintain efficient training, $\mathcal{S}$ must be easily and effectively encoded into features.

Considering the first and second points, $\mathcal{S}$ should be in an explicit representation. Further considering the third point, the main explicit 3D representations that can be easily encoded as features are voxels and point clouds. Both representations are suitable, but voxels typically require a high resolution to accurately represent shapes, and processing high-resolution voxels into features is computationally expensive. Additionally, voxels, being a discrete representation, are less precise for data augmentation compared to point clouds. Therefore, we chose point clouds as the representation for $\mathcal{S}$. To enhance the expressive power of the point clouds, we also include normals into the point cloud representation.

To obtain point clouds from the ground truth mesh for training, we could simply sample point clouds directly from the surface of $\mathcal{M}$. However, this would create problems during inference: the surfaces of automatically generated 3D assets are often rougher than those of AMs. For example, in AMs, we would sample a series of points on a flat plane, whereas automatically generated 3D assets would have uneven surfaces, causing a domain gap between training and inference.

Therefore, we need to ensure that $\mathcal{S}$ extracted from the ground truth $\mathcal{M}$ during training has a similar domain to the $\mathcal{S}$ extracted during inference. To bring their domains closer, we intentionally construct coarse meshes from AMs. We first extract the signed distance function from $\mathcal{M}$ with (Wang et al., 2022), then convert it into a relatively coarse mesh using Marching Cubes (Lorensen & Cline, 1987) to destroy the ground truth topology. Finally, we sample point cloud and its normals from the coarse mesh. This approach also helps to avoid overfitting, as AMs typically have fewer faces, and each face can often sample multiple points. The network can easily recognize the ground truth topology by determining whether the points lie on the same plane.

Since almost all 3D representations can be converted into a coarse mesh using Marching Cubes (Lorensen & Cline, 1987) or sampled into point clouds, this ensures that the domain of $\mathcal{S}$ is consistent during both training and inference. We pair the point clouds extracted as $\mathcal{S}$ with $\mathcal{M}$ to create a data item $\{(\mathcal{M}_i, \mathcal{S}_i)\}_i$ for training.

## 4.2 VQ-VAE with Noise-Resistant Decoder

Following MeshGPT (Siddiqui et al., 2023), we first train a VQ-VAE (Van Den Oord et al., 2017) to learn a vocabulary of geometric embeddings for better transformer (Vaswani et al., 2017) learning. Different to MeshGPT, which uses graph convolutional networks (Wu et al., 2019) and ResNet (He et al., 2016) as the encoder and decoder respectively, we employ transformers with identical structures for both the encoder and decoder. When training VQ-VAE, meshes are discretized and input as a sequence of triangle faces:

$$\mathcal{M} := (f_1, f_2, f_3, \ldots, f_N), \tag{5}$$

where $f_i$ is the coordinates of the vertices of each face, and $N$ is the number of faces in $\mathcal{M}$. The encoder $E$ then extracts a feature vector for each face:

$$\mathcal{Z} = (z_1, z_2, \ldots, z_N) = E(\mathcal{M}), \tag{6}$$

where $z_i$ is the feature vector for $f_i$.

The extracted faces are then quantized into quantized features $\mathcal{T}$ with codebook $\mathcal{B}$:

$$\mathcal{T} = RQ(\mathcal{Z}; \mathcal{B}) \tag{7}$$

Finally, the reconstructed mesh is decoded from $\mathcal{T}$ with decoder $D$ by predicting the logits for each vertex's coordinates:

$$\hat{\mathcal{M}} = D(\mathcal{Z}) \tag{8}$$

The VQ-VAE is trained end-to-end with cross-entropy loss on the predicted vertex coordinate logits and the commitment loss of vector quantization (Van Den Oord et al., 2017). After the training of VQ-VAE, the encoder-decoder of VQ-VAE is treated as a tokenizer and detokenizer for autoregressive transformer training.

However, as shown in Fig. 7, there are possible imperfections in the generation results. To address this issue, given our setting of Shape-Conditioned AM Generation, the VQ-VAE decoder can also take the shape condition as input. Small imperfections in the token sequences generated by the transformer can potentially be corrected by a shape-aware decoder. Therefore, after completing the vanilla VQ-VAE training, we add an additional decoder fine-tuning stage, where we inject the shape information into the transformer decoder. Then we add random Gumbel noise to the codebook sampling logits to simulate the potential imperfections in the token sequences generated by the transformer during inference. The decoder is then updated independently with the same cross-entropy loss to train it to produce refined meshes even when facing imperfect token sequences. Our experiments in Tab. 3 and Tab. 4 show that our method effectively enhances the decoder's noise resistance and mesh generation quality.

## 4.3 Shape-Conditioned Autoregressive Transformer

To add shape condition to the transformer, inspired by the success of multimodal large language models (Wu et al., 2023; Liu et al., 2024b; Xu et al., 2023; Guo et al., 2023), we first encode the point cloud into a fixed-length token sequence with a point cloud encoder $\mathcal{P}$ and then concatenate it to the front of the embedding sequence from $\mathcal{T}$ VQ-VAE as the final input embedding sequence for the transformer:

$$\mathcal{T}' = \text{concat}(\mathcal{P}(\mathcal{S}), \mathcal{T}) \tag{9}$$

where $\mathcal{T}'$ is the training input for the transformer.

We borrow a pretrained point encoder from (Zhao et al., 2024) and add a linear projection layer to project its output feature to the same latent space as $\mathcal{T}$. During training, the original point encoder from (Zhao et al., 2024) is frozen; we only update the newly added projection layer and the autoregressive transformer with cross-entropy loss.

During inference, we input $\mathcal{P}(\mathcal{S})$ to the transformer and require it to generate the subsequent sequence, $\hat{\mathcal{T}}$. $\hat{\mathcal{T}}$ is then input to the noise-resistant decoder to reconstruct meshes:

$$\hat{\mathcal{M}} = D(\hat{\mathcal{T}}) \tag{10}$$

where $\hat{\mathcal{M}}$ is the final generated AM.

Table 1: Comparison of Mesh Generation Methods. As shown in the left table, compared to the baseline Artist-Created Mesh Generation method, the meshes generated by MeshAnything are better aligned with human preferences. In the right table, we compare MeshAnything with mesh extraction baselines, and it received the most votes. For detailed settings, please refer to Section 5.4.

| Method | Shape↑ | Topology↑ | Method | Shape↑ | Topology↑ |
|---|---|---|---|---|---|
| PolyGen | 12.7% | 11.1% | MarchingCubes | 38.1% | 10.2% |
| MeshGPT | 24.1% | 28.2% | Shape As Points | 17.3% | 6.2% |
| MeshAnything | **63.2%** | **60.7%** | MeshAnything | **44.6%** | **83.6%** |

We use the standard next-token prediction loss to train shape-conditioned transformers. For each sequence, we add a `<bos>` token after the point cloud tokens and a `<eos>` token after the mesh tokens to identify the end of a 3D mesh.

## 5 EXPERIMENTS

### 5.1 DATA PREPARATION

**Data Selection.** Existing AM generation works are limited to a few categories. However, our method targets to operate on general shapes. MeshAnything is trained on a combined dataset of Objaverse (Deitke et al., 2023b) and ShapeNet (Chang et al., 2015), selected for their complementary characteristics. We chose Objaverse because it contains a large number of AMs without category limitations. On the other hand, ShapeNet offers higher data quality within limited categories.

We filter out meshes with more than 800 faces from both datasets. Additionally, we manually filtered out low quality meshes. Our final filtered dataset consists of 51k meshes from Objaverse and 5k meshes from ShapeNet. We randomly select 10% of this dataset as the evaluation dataset, with the remaining 90% used as the training set for all our experiments.

**Data Processing and Augmentation.** Following the strategies of PolyGen (Nash et al., 2020) and MeshGPT (Siddiqui et al., 2023), we order faces by their lowest vertex index, then by the next lowest, and so on. Vertices are sorted in ascending order based on their z-y-x coordinates, where z represents the vertical axis. Within each face, we permute the indices to ensure the lowest index comes first. During training, we apply on-the-fly scaling, shifting, and rotation augmentations, normalizing each mesh to a unit bounding box from $-0.5$ to $0.5$.

### 5.2 IMPLEMENTATION DETAILS

The encoder and decoder of VQ-VAE both use the encoder of BERT (Devlin et al., 2018), while we choose OPT-350M (Zhang et al., 2022) as our autoregressive transformer architecture. The residual vector quantization (Zeghidour et al., 2021) depth is set to 3, with a codebook size of 8,192.

Our point encoder is based on the pretrained point encoder from (Zhao et al., 2024), which has been trained on Objaverse and thus can handle general shapes. This point encoder outputs a fixed-length token sequence of 257 tokens, with 256 tokens primarily containing shape information and an additional head token containing semantic information about the shape. We sample 4096 points for each point cloud.

The training batch size for both the VQ-VAE and the transformer is set to 8 per GPU. The VQ-VAE is trained on 8 A100 GPUs for 12 hours, after which we separately finetune the decoder part of the VQ-VAE into a noise-resistant decoder, as detailed in Section 4.2. Following this, the transformer is trained on 8 A100 GPUs for 4 days.

### 5.3 QUALITATIVE EXPERIMENTS

As shown in Fig. 1, MeshAnything effectively generates AMs from various 3D representations. In our experiments, we use Rodin (Team, 2024) as the text-to-3D and image-to-3D method, and employ (Mildenhall et al., 2020) and (Kerbl et al., 2023a) as the 3D reconstruction pipeline to obtain

Table 2: **Quantitative Comparisons with Prior Arts on Objaverse.** MeshAnything significantly outperforms prior methods across all metrics. MMD, KID are scaled by $10^3$.

| Method | COV↑ | MMD↓ | 1-NNA↓ | FID↓ | KID↓ |
|---|---|---|---|---|---|
| PolyGen | 23.2 | 6.22 | 88.2 | 48.8 | 27.7 |
| MeshGPT | 41.7 | 3.83 | 67.3 | 25.1 | 6.11 |
| MeshAnything | **53.1** | **2.72** | **55.7** | **14.5** | **1.89** |

the corresponding NeRF and Gaussian Splatting models. For additional qualitative results, please refer to A.2 combined with other 3D asset production pipelines.

## 5.4 QUANTITATIVE EXPERIMENTS

From the generative model perspective, MeshAnything is a shape-conditioned mesh generation model. From the mesh extraction perspective, it extracts artist-created meshes from point clouds. Consequently, we compare MeshAnything with both types of methods. Additional experiments can be found in Appendix Section A.2.

**User Study.** As shown in Tab. 1, we conducted two user studies, comparing with mesh generation baselines (Nash et al., 2020; Siddiqui et al., 2023) and mesh extraction baselines (Lorensen & Cline, 1987; Peng et al., 2021), respectively. The majority of participants in our user study were researchers from the 3D field, with a smaller portion being 3D industry practitioners. The mesh generation baselines are trained on ShapeNet, and to ensure a fair comparison, we retrained them on Objaverse using the same transformer model as MeshAnything. Since the mesh generation baselines are all unconditional mesh generation methods, whereas MeshAnything is a shape-conditioned mesh generation method, we sampled shapes randomly from the evaluation set of Objaverse as inputs for MeshAnything, while for the baseline methods, we performed random sampling directly.

In the mesh extraction baseline, since our method can also be viewed as a point cloud to mesh approach, we included (Peng et al., 2021), a point cloud to mesh method, as a baseline. Additionally, we optimized the results from the mesh extraction baseline using the Blender remesh method (Blender Development Team, 2024) to simplify the topology.

We collected 30 results from each method and asked users to vote for the best one in terms of shape quality and topology quality. A total of 41 users participated, providing 1,230 valid comparisons. Both user studies demonstrated the superiority of our method. The only difference between the retrained MeshGPT and MeshAnything is whether they are shape-conditioned, further proving the advantages of the shape-conditioned mesh generation setting.

**Metrics.** We follow the metric setting of (Chen et al., 2022; Siddiqui et al., 2023). We detail this setting in Appendix Section. A.1.

**Comparison with Mesh Generation Pipelines.** We use the same retrained models from the user study for comparison. As shown in Tab. 2, MeshAnything significantly outperforms prior methods (Nash et al., 2020; Siddiqui et al., 2023), indicating that it's superior in both the shape and topology quality. Since the only difference between the retrained MeshGPT and MeshAnything is the inclusion of shape conditioning, the superior performance of MeshAnything further demonstrates that Shape-Conditioned Mesh Generation is a more suitable setting for mesh generation.

## 6 CONCLUSION

In this work, we propose a novel setting for improved mesh extraction and mesh generation, namely Shape-Conditioned Artist-Created Mesh (AM) Generation. Following this setting, we introduce MeshAnything, a model capable of generating AMs that adhere to given 3D assets. MeshAnything can convert 3D assets in any 3D representation into AMs and thus can be integrated with diverse 3D asset production methods to facilitate their application in the 3D industry. Extensive experiments demonstrate the superior performance of our method, highlighting its potential to scale up for 3D industry application and its advantage over previous methods.

## 7 ACKNOWLEDGEMENT

This study is supported under the RIE2020 Industry Alignment Fund – Industry Collaboration Projects (IAF-ICP) Funding Initiative, as well as cash and in-kind contribution from the industry partner(s). This research is also supported by the MoE AcRF Tier 2 grant (MOE-T2EP20223-0001).

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

# A APPENDIX

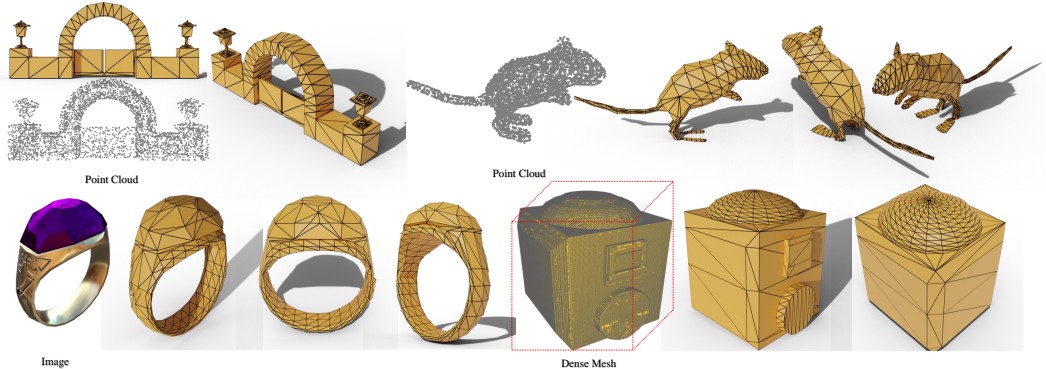

Figure 5: **Additional qualitative results of MeshAnything.** As shown, MeshAnything can be integrated with various 3D production pipelines to achieve highly controllable mesh generation.

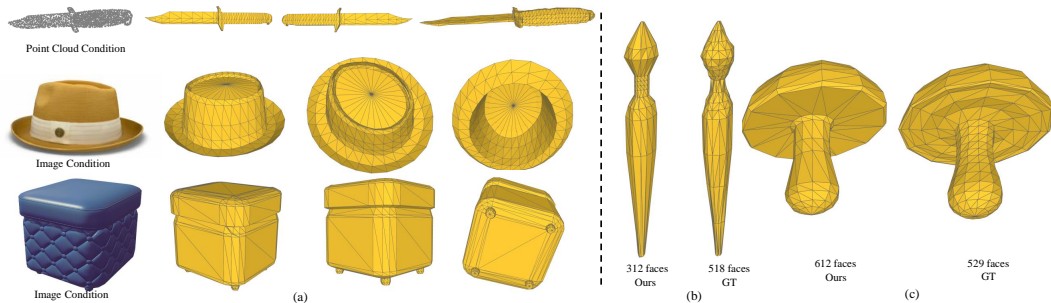

Figure 6: **Qualitative Results.** (a) further demonstrates our capability to achieve highly controllable mesh generation when combined with 3D asset production pipelines. Besides, we compare our reseults with ground truth in (b) and (c). In (b), MeshAnything generates meshes with better topology and fewer faces than the ground truth. In (c), we produce meshes with a completely different topology while achieving a similar shape, proving that our method does not simply overfit but understands how to construct meshes using efficient topology.

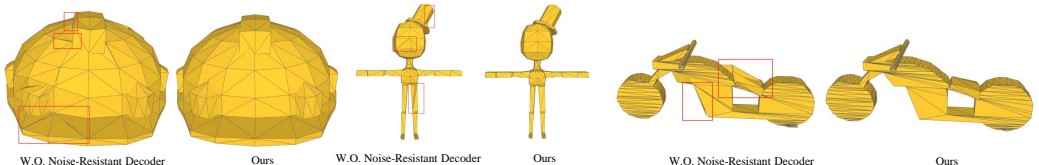

Figure 7: **Ablation on Noise-Resistant Decoder.** The decoder-only transformer may generate low-quality token sequences, and the decoder of VQ-VAE would typically produce flawed meshes based on these sequences. In contrast, our Noise-Resistant Decoder, aided by shape conditions, has the ability to resist these low-quality token sequences, producing higher-quality meshes.

## A.1 METRICS

We follow the evaluation metric setting of (Siddiqui et al., 2023) in mesh generation experiments and the setting of (Chen et al., 2022) in mesh extraction experiments.

We quantitatively evaluate mesh quality by uniformly sampling 100K points from the faces of both the ground truth meshes and the predicted meshes, and then computing a set of metrics to assess various aspects of the reconstruction.

Table 3: Reconstruction Performance under Different Noise Levels with and without Noise-Resistant (NR) Decoder. Please refer to A.1 for metrics explanation.

| Noise Level | CD($\times 10^{-2}$)↓ | | ECD($\times 10^{-2}$)↓ | | NC↑ | |
|---|---|---|---|---|---|---|
| | W/O NR | W/ NR | W/O NR | W/ NR | W/O NR | W/ NR |
| 0.0 | 0.011 | **0.007** | 0.035 | **0.023** | 0.987 | **0.993** |
| 0.1 | 0.187 | **0.028** | 0.613 | **0.138** | 0.973 | **0.991** |
| 0.5 | 1.167 | **0.639** | 2.538 | **1.329** | 0.964 | **0.981** |
| 1.0 | 2.131 | **1.798** | 4.317 | **2.316** | 0.952 | **0.969** |

Table 4: Ablation on Noise-Resistant (NR) Decoder for the Quality of Mesh Generation.

| Method | CD↓ ($\times 10^{-2}$) | ECD↓ ($\times 10^{-2}$) | NC↑ |
|---|---|---|---|
| W/O NR | 2.423 | 6.414 | 0.883 |
| W/ NR | **2.256** | **6.245** | **0.902** |

For mesh extraction, we report the following metrics: Chamfer Distance (CD) to evaluate the overall quality of a reconstructed mesh; Edge Chamfer Distance (ECD) to assess the preservation of sharp edges by sampling points near sharp edges and corners; and Normal Consistency (NC) to evaluate the quality of the surface normals. Additionally, we report the number of mesh vertices (#V) and the number of mesh faces (#F). We also provide the ratio of the estimated number of vertices to the ground truth number of vertices (#V_R) and the same ratio for faces (#F_R).

For mesh generation, Coverage (COV) captures the diversity of generated meshes and is sensitive to mode dropping, but it does not reflect the quality of the results. Minimum Matching Distance (MMD) measures the average distance between the reference set and their nearest neighbors in the generated set, though it lacks sensitivity to low-quality outputs. The 1-Nearest Neighbor Accuracy (1-NNA) assesses both quality and diversity between the generated and reference sets. To evaluate topology quality, we render the ground truth meshes and generated meshes with their wireframes visualized. We then employ Frechet Inception Distance (FID) and Kernel Inception Distance (KID) on rendered images. MMD, and KID scores are scaled by a factor of $10^3$.

## A.2 EXPERIMENTS

**Additional Qualitative Experiments** We present more qualitative results of MeshAnything here. As shown in Fig. 5 and Fig. 6, MeshAnything effectively generates AMs from various 3D representations. When integrated with different 3D assets production pipelines, our method effectively achieves mesh generation with diverse conditions.

Next, Fig. 6 demonstrates that MeshAnything does not simply overfit but understands how to generate meshes with efficient topology that conform to the given shape. To prove this, we use manually-created meshes as ground truth and use their shapes as conditions to test whether our model can generate meshes with comparable topology. To effectively use the ground truth as conditions, we first convert them into dense meshes using Marching Cubes (Lorensen & Cline, 1987) to disrupt their face structure. Then, we sample point clouds with normals from the dense meshes to serve as shape conditions. The experimental results in Fig. 6 show that MeshAnything is capable of generating meshes comparable to or even surpassing those modeled by human artists, exhibiting diverse and strong 3D modeling capabilities.

**Comparison with mesh extraction baselines.** Our method is related to various mesh extraction methods (Lorensen & Cline, 1987; Chen & Zhang, 2021; Chen et al., 2022; Shen et al., 2023; Peng et al., 2021) since we also convert other 3D representations into meshes. However, it is important to note that previous approaches are reconstruction-like methods that produce dense meshes, while our approach is generative, creating Artist-Created Meshes (AMs) that are significantly more complex to produce than dense meshes. Therefore, strictly speaking, our method cannot be considered the same as these reconstruction-based mesh extraction methods. The main purpose of this comparison

Table 5: Quantitative evaluation with mesh extraction baselines. MC, FC, SAP refer to Marching Cubes Lorensen & Cline (1987), FlexiCubes Shen et al. (2023), and Shape As Points Peng et al. (2021), respectively. Please refer to A.1 for metrics explanation.

| Method | CD↓ $(\times 10^{-2})$ | ECD↓ $(\times 10^{-2})$ | NC↑ | #V↓ $(\times 10^3)$ | #F↓ $(\times 10^3)$ | V_R↓ | F_R↓ |
|---|---|---|---|---|---|---|---|
| (a) Marching Cubes | 1.532 | 6.733 | 0.954 | 73.22 | 146.0 | 440.2 | 462.2 |
| (b) MC+Remesh (0.005) | 2.174 | 7.813 | 0.912 | 127.8 | 167.9 | 748.1 | 534.6 |
| (c) MC+Remesh (0.010) | 2.083 | 7.578 | 0.929 | 39.01 | 41.78 | 225.4 | 132.3 |
| (d) MC+Remesh (0.030) | 2.915 | 8.329 | 0.863 | 5.848 | 4.410 | 34.38 | 14.05 |
| (e) MC+Remesh (0.050) | 4.179 | 8.138 | 0.814 | 2.299 | 1.538 | 13.64 | 4.920 |
| (f) MC+Remesh (0.100) | 7.312 | 10.771 | 0.748 | 0.625 | 0.359 | 3.735 | 1.149 |
| (g) FC | **1.190** | **6.121** | **0.967** | 59.12 | 121.1 | 378.2 | 391.1 |
| (h) FC+Remesh (0.010) | 1.861 | 6.940 | 0.933 | 37.98 | 40.19 | 205.5 | 124.2 |
| (i) SAP | 1.771 | 7.112 | 0.939 | 79.12 | 152.3 | 481.2 | 489.3 |
| (j) SAP+Remesh (0.010) | 2.367 | 7.862 | 0.925 | 39.17 | 42.87 | 239.1 | 136.6 |
| (k) *MeshAnything* | 2.256 | 6.245 | 0.902 | **0.172** | **0.318** | **0.888** | **0.871** |

is to use these mesh extraction methods as a reference for evaluating the quality of the meshes generated by MeshAnything in terms of shape. We compare MeshAnything with Lorensen & Cline (1987); Shen et al. (2023); Peng et al. (2021). Among these, MarchingCubes is the most popular mesh extraction method, FlexiCubes represents the state-of-the-art in mesh extraction, and Shape as Points is the leading method for extracting mesh from point cloud.

We also combined these methods with the remesh technique to test whether they could significantly reduce the number of faces while maintaining shape quality. We used Blender Remesh in voxel mode (Community, 2018; Blender Development Team, 2024), specifically using Blender version 4.1, as the remesh method. Since our evaluation dataset includes non-watertight meshes, we first extract the signed distance fields (SDF) of all ground truth meshes using (Wang et al., 2022), which can handle non-watertight meshes. We then apply Marching Cubes with a resolution of 128 on these SDFs. Next, we apply Blender remesh (Blender Development Team, 2024) with different voxel sizes to the Marching Cubes results, as both the remesh method and our approach are capable of simplifying topology. Additionally, the Marching Cubes result is used as the shape condition input to MeshAnything to obtain our results. The settings of (Shen et al., 2023) and (Peng et al., 2021) follow their papers.

As shown in Tab. 5, we found that these methods require hundreds of times more faces to achieve results comparable to our method. Comparing (a), (g), (i) and (k), our method lags in Chamfer Distance (CD) and Normal Consistency (NC), mainly due to our method's inherent failure cases as a generative model, which makes it less robust than these reconstruction-based mesh extraction methods. When comparing with remesh methods, we observe that they incur a high cost to achieve a face count similar to ours. Comparing (f) and (k), we find that even when remesh methods achieve a comparable face count, the number of vertices is still several times higher than ours, indicating that the topology efficiency of remesh methods is far inferior to ours, as they completely ignore the shape characteristics of the 3D assets. It's important to note that the metrics in mesh etraction can only indicate the quality of shape alignment, which do not effectively reflect the topological advantages of our method. Additionally, we surprisingly find that our method can produce results with fewer faces than the ground truth, demonstrating that MeshAnything is not overfitting to the data but instead learns an efficient topology representation, occasionally surpassing the ground truth meshes.

**Ablations on Noise-Resistant Conditional Decoder.** We perform ablation experiments to verify the effectiveness of the Noise-Resistant Decoder. We begin with a VQ-VAE trained without any noise or conditioning. We then perform ablation between two settings: one where the decoder remains unchanged and unaware of the shape condition, and another where the shape condition is injected into the transformer, as described in Section 4.2. Next, we randomly sample a noise from gumbel distribution and add it to codebook sampling logits during the vector quantization process to simulate the potential low-quality token sequences generated by the transformer. We control the noise level by scaling the added noise.

Table 6: Experiments on the Impact of Input Point Cloud Quality on Generated Results.

| Method | CD↓ $(\times 10^{-2})$ | ECD↓ $(\times 10^{-2})$ | NC↑ | #V↓ $(\times 10^3)$ | #F↓ $(\times 10^3)$ | V_R↓ | F_R↓ |
|---|---|---|---|---|---|---|---|
| (a) Noise scale 0.005 | 2.351 | 6.412 | 0.897 | 0.175 | 0.321 | 0.895 | 0.880 |
| (b) Noise scale 0.020 | 2.980 | 6.970 | 0.881 | 0.180 | 0.330 | 0.901 | 0.910 |
| (c) Noise scale 0.050 | 4.910 | 8.556 | 0.755 | **0.162** | **0.284** | **0.811** | **0.802** |
| (d) Rodin | 2.552 | 6.622 | 0.833 | 0.185 | 0.342 | 0.919 | 0.923 |
| (e) *MeshAnything* | **2.256** | **6.245** | **0.902** | 0.172 | 0.318 | 0.888 | 0.871 |

Table 7: **The comparison experiment between MeshAnything and MeshGPT on the chair category in ShapeNet**. This experiment is conducted with the same settings as in Table 2, except that both training and evaluation were performed exclusively on the chair category in ShapeNet.

| Method | COV↑ | MMD↓ | 1-NNA↓ | FID↓ | KID↓ |
|---|---|---|---|---|---|
| MeshGPT | 49.2 | 2.98 | 61.0 | 16.4 | 2.04 |
| MeshAnything | **62.1** | **1.92** | **49.8** | **11.2** | **1.21** |

After training both models for enough epochs, we test their performance to the same level of noise. As shown in Tab. 3, as the intensity of the added noise increases, the Noise-Resistant Decoder with shape condition clearly achieves better reconstruction results. This indicates that the shape condition helps the decoder identify and correct imperfections in the input token sequences.

Next, we verify whether the Noise-Resistant Decoder indeed enhances the transformer's performance during inference. The test method used dense meshes derived from corrupted GT meshes as the condition for generating new meshes. The generated meshes were then assessed for shape alignment with the conditional shape. As shown in Tab. 4, the model with Noise-Resistant Decoder achieved better results.

**Experiments on the Impact of Input Point Cloud Quality on Generated Results.** MeshAnything takes point clouds as input, and its robustness to point cloud quality determines its versatility across various applications. We design two experiments to evaluate its tolerance to input point cloud quality: First, keeping the other evaluation settings unchanged, we apply Gaussian noise to the input point cloud coordinates and normals. Specifically, for each point, we randomly sample Gaussian noise from a standard distribution, scale it by a noise factor, and add it to the point's coordinates. The same approach is applied to the normals, but normalization is applied after adding the noise. Second, we use Rodin's generation result as the ground truth mesh, sample point clouds from this mesh as input, and evaluate the deviation between the generated result and the ground truth.

As shown in Tab. 6, MeshAnything did not experience a significant performance drop in (a) and (b), demonstrating resilience to noise in the point cloud, with a noticeable performance decrease only in (c). It is important to note that the input point cloud is normalized to the range [-1,1], and the noise scale in (c) is already quite large. The experiment in (d) further demonstrates that MeshAnything can tolerate generated point clouds and effectively integrate with 3D generation models.

**Experiments between MeshAnything and MeshGPT on the chair category in ShapeNet.** To reduce the complexity of learning the shape distribution, we conducted the comparison experiment between MeshAnything and MeshGPT (Siddiqui et al., 2023) exclusively on the chair category in ShapeNet. As shown in 7, the gap between MeshGPT and MeshAnything narrowed under this setting, though a noticeable difference remains. This result supports our argument in Sec. 3 that shape-conditioned mesh generation is considerably less challenging to learn than unconditional mesh generation.

**Sampling Strategy and Diversity of Generated Results** All of our qualitative and quantitative experiments employed multinomial sampling, which is equivalent to a beam search with a beam size of 1. We used top_k and top_p values set to 50 and 0.95, respectively. We provide the generation variants obtained using this sampling approach in Fig. 8. As shown in the figure, MeshAnything demonstrates the capability to generate various topologies.

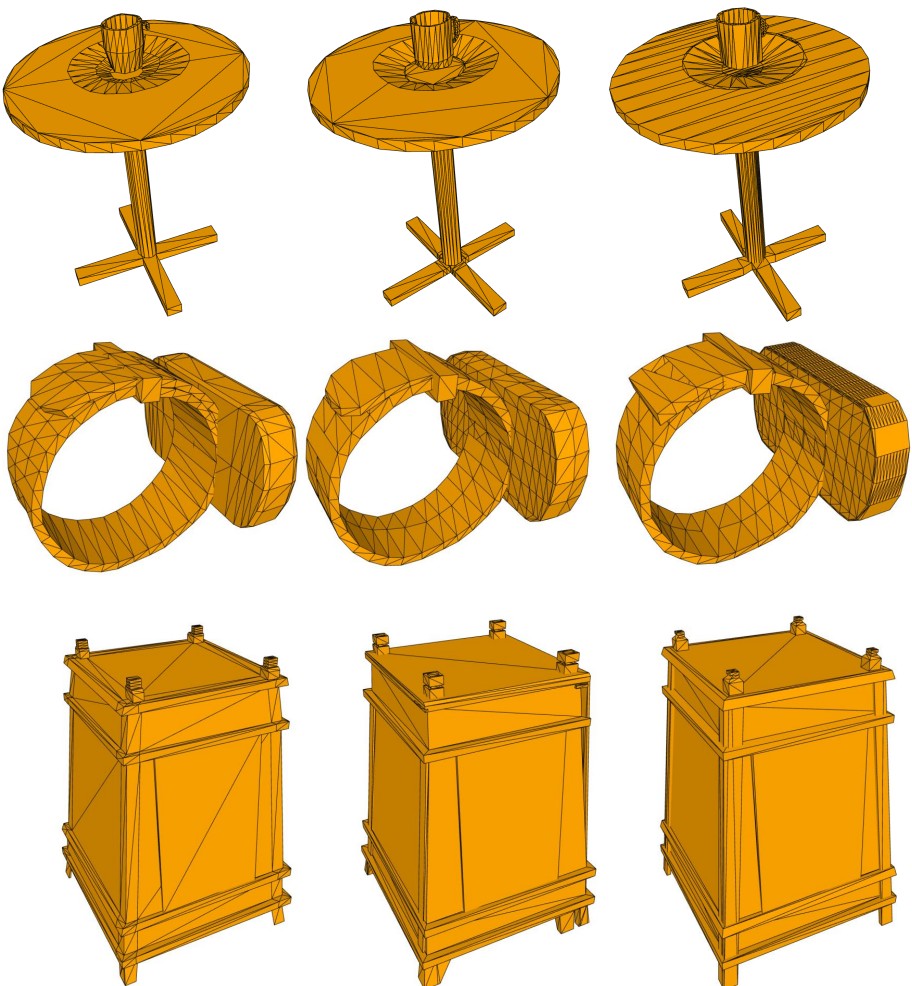

Figure 8: **Diversity of Generated Results.** The generated results of MeshAnything obtained using multinomial sampling with different random seeds.

**Qualitative Comparison with MeshGPT**. We provide a quantitative comparison with MeshGPT (Siddiqui et al., 2023) and Polygen (Nash et al., 2020). As shown in Fig. 9, it is challenging for these two methods to produce high-quality meshes on Objaverse. This advantage is primarily because MeshAnything avoids the need to learn the complex 3D shape distribution, making the training process significantly easier compared to unconditional mesh generation.

### A.3    LIMITATIONS

Our method cannot generate meshes that exceed the maximum face count limit, which restricts its ability to convert large scenes and highly complex objects into meshes. Additionally, due to its generative nature, our method is not as stable as reconstruction-based mesh extraction methods like (Lorensen & Cline, 1987; Shen et al., 2023). After decades of development, reconstruction methods achieve near 100% success rates with reasonable inputs. However, as a generative approach, our method inevitably produces occasional failure cases.

### A.4    FUTURE WORKS.

Autoregressive mesh generation remains inefficient, significantly limiting the practical application of this approach. Developing more efficient representations for mesh generation is essential. Fur-

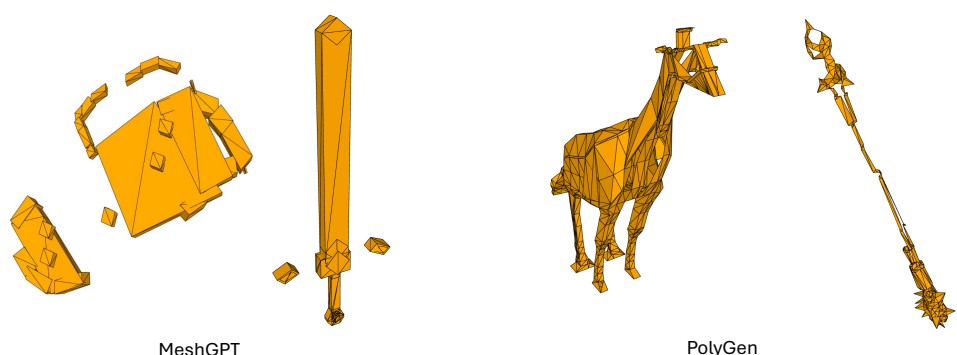

MeshGPT                                              PolyGen

Figure 9: **Qualitative Comparison with MeshGPT (Siddiqui et al., 2023) and PolyGen (Nash et al., 2020).** Since MeshAnything is conditioned on point cloud input, a direct qualitative comparison with these two methods is not feasible. Therefore, we showcase unconditional generation results from MeshGPT and PolyGen. For a fair comparison, both methods use the same architecture and data as our approach. As shown, when scaled to Objaverse, MeshGPT and PolyGen struggle to consistently produce meshes with complex, general shapes.

thermore, improving the success rate of mesh generation is critical. Due to the cumulative error characteristic of autoregressive models, a deviation in any one token prediction can cause the entire mesh generation process to fail. Future developments in this line of research may benefit from techniques used in large language models (LLMs) to mitigate similar issues. Additionally, current mesh generation methods are limited to the object level. Extending these methods to the scene level is equally important, as it would greatly expand the range of potential applications.

## A.5 SOCIAL IMPACT

Our method points to a promising approach for the automatically generation of Artist-Created Meshes, which has the potential to significantly reduce labor costs in the 3D industry, thereby facilitating advancements in industries such as gaming, film, and the metaverse. However, the reduced cost of obtaining 3D Artist-Created meshes could also lead to potential criminal activities.

