# OpenReview forum: "MeshAnything: Artist-Created Mesh Generation with Autoregressive Transformers"
_ICLR.cc/2025/Conference — ICLR 2025 Poster_

### Official Review · Reviewer_3Tqd · 2024-10-26

**Soundness:** 3
**Presentation:** 4
**Contribution:** 4
**Rating:** 8
**Confidence:** 4

**Summary:**

This paper presents MeshAnything, a model framing mesh extraction as a generative task, resulting in artist-crafted meshes that align with specified shapes. The approach first establishes a mesh vocabulary through VQ-VAE, followed by training a shape-conditioned, decoder-only transformer with a noise-resistant decoder design on the learned vocabulary, enabling shape-conditioned autoregressive mesh generation. This paper is well-crafted, and I anticipate that it will positively influence the field of 3D content generation though the pipeline appears somewhat straightforward. The performance of the method seems promising.

**Strengths:**

This is a well-written paper that clearly conveys the authors' contributions. The topic is highly relevant to the field, as it addresses the gap between generated shapes and practical applications - I expect the authors will consider open-sourcing their code to facilitate further research. The effectiveness of the method is demonstrated in the experimental sections.

**Weaknesses:**

I recommend that the authors showcase real-world application examples to highlight the significance of their research. For example, the authors could demonstrate how Artist-Created Meshes (AMs) improve shape manipulation for artist-driven shape modifications or offer enhanced rendering performance (e.g., efficiency) compared to conventional dense meshes.

## Related Works
Additionally, recent advancements in 3D reconstruction and generation could be acknowledged, such as:
- Part123: Part-aware 3D Reconstruction from a Single-view Image (SIGGRAPH 2024)
- SyncDreamer: Generating Multiview-consistent Images from a Single-view Image ICLR 2024.
- CLAY: A Controllable Large-scale Generative Model for Creating High-quality 3D Assets (SIGGRAPH 2024)


## Exposition

- "MeshAnything converts any 3D representation into Artist-Created Meshes (AMs), i.e., meshes created by human artists." Consider rephrasing "any 3D representation" to "different 3D representations."

- "This is because all meshes produced by these methods (Lorensen & Cline, 1987; Chernyaev, 1995; Lorensen & Cline, 1998; Shen et al., 2021b; Chen et al., 2022; Shen et al., 2023) exhibit significantly poorer topology quality compared to AMs. As shown in Fig. 2, these methods rely on dense faces to reconstruct 3D shapes, completely ignoring geometric characteristics." The logic here seems unclear. Generally, mesh topology pertains to the connectivity of vertices, while geometric features relate to vertex positions. Could the authors clarify how "poorer topology quality (relying on dense faces)" affects "geometric characteristics"?

**Questions:**

1. Will the data and code be available for reproducibility?
2. Can the limitations and potential future work be expanded in the main paper? Are there any failure cases the method cannot handle? Additionally, what would happen if watertightness is required during generation? Including these discussions could enhance the paper.
3. "Due to its generative nature, our method is not as stable as reconstruction-based mesh extraction methods." Could the authors clarify the meaning of "stable" in this context?
4. How does the method perform on open surfaces?

---

> ### Author Response · Authors · 2024-11-17
> **Response to reviewer 3Tqd**
>
> We thank reviewer 3Tqd for the insightful comments and constructive feedback. We will add real-world application examples in the main text to highlight the significance of our research. Our responses to the specific questions are as follows:
>
> ### 1. Related Works
>
> Thank you for the recommendation. We have added these related works to our updated paper in Page 5, line 234.
>
> ### 2. Exposition
>
> We appreciate the suggestions and have made the recommended edits in the main text.
>
> To clarify how "poorer topology quality affects geometric characteristics," this statement is intended to convey that topology quality also impacts shape quality. Reconstruction methods, such as Marching Cubes, inherently lack the ability to represent precise geometry (such as sharp angles at specific degrees), often relying on dense faces for approximation, which limits their ability to produce sharp geometric details.
>
> ### 3. Data and Code Availability (Responding to R4Q1)
>
> Of course, we will release all our code and data upon paper acceptance.
>
> ### 4. Limitations and Future Work (Responding to R4Q2)
>
> Thank you for the suggestion. We have expanded the discussion on limitations and future work in our updated paper (from Page 20, line 1068 to line 1106). Generating highly complex meshes with many surfaces remains challenging due to the high face count required. However, we are pleased to see recent advances that successfully scale up our method [1,2,3], enabling effective generation of such complex meshes.
>
> Ensuring watertightness during generation can be achieved by incorporating a simple prompt into the training process. As approximately 50% of meshes in Objaverse are watertight, training a model capable of generating watertight meshes upon request is feasible.
>
> ### 5. Stability Clarification (Responding to R4Q3)
>
> In this context, "stability" refers to the success rate during inference. After decades of development, reconstruction methods achieve near 100% success rates with reasonable inputs. However, as a generative approach, our method inevitably produces occasional failure cases.
>
> ### 6. Performance on Open Surfaces
>
> For an example of our method’s performance on open surfaces, please see the second mesh in the first row of Figure 1, which depicts a mesh with open surfaces. Conceptually, our approach can handle both open and closed surfaces equally well, and open surfaces may be simpler for our method to generate due to the typically lower face count required.
>
> ---
>
> ### References
> 1. Edgerunner: Auto-regressive Auto-encoder for Artistic Mesh Generation. Tang et al., 2024. [https://arxiv.org/abs/2409.18114](https://arxiv.org/abs/2409.18114)
> 2. Meshtron: High-Fidelity, Artist-Like 3D Mesh Generation at Scale. Anonymous, 2024. [https://openreview.net/forum?id=mhzDv7UAMu](https://openreview.net/forum?id=mhzDv7UAMu) [https://meshtron.github.io/index.html](https://meshtron.github.io/index.html)
> 3. Scaling Mesh Generation via Compressive Tokenization. Weng et al., 2024. [https://arxiv.org/abs/2411.07025](https://arxiv.org/abs/2411.07025)

---

> > ### Comment · Reviewer_3Tqd · 2024-11-21
> > **Reply to the authors**
> >
> > Thank you for your response and the updated submission. The explanation has greatly clarified the work for me. I encourage you to incorporate the promised discussion and results in your revision. For example, including visualizations and analyses of failure cases (the required watertightness is not achieved) could provide readers with a deeper understanding of the method.
> >
> > > Conceptually, our approach can handle both open and closed surfaces equally well, and open surfaces may be simpler for our method to generate due to the typically lower face count required.
> >
> > On a minor note, I believe this property of producing arbitrary topologies could be emphasized more in the writing as a significant advantage of this explicit modeling approach. Also, this needs to be differentiated between the surface topology.
> >
> > Overall, I still lean towards the positive of this work for Artist-Created Mesh Generation and partially its performance, as this could potentially play a role in bridging the gap between AI-generated 3D content and real-world applications, especially in modeling. The authors also promised to release the code and data. However, I understand the concerns raised by other reviewers, particularly regarding "artist-generated meshes" and their scalability. I believe these aspects should be elaborated on further to strengthen the work.

---

### Official Review · Reviewer_qFPE · 2024-11-02

**Soundness:** 3
**Presentation:** 3
**Contribution:** 3
**Rating:** 6
**Confidence:** 5

**Summary:**

This paper tackles the important question of converting various 3D representations to explicit mesh representations similar to what artists create. For this, the authors adopt the idea of MeshGPT and inject shape-conditioned information to make the model focus on learning topology. To make the model agnostic to specific 3D representations, e.g., NeRF or Gaussian Splats, the authors choose to convert all representations to point clouds and generate the mesh based on the features from the point cloud. Further, the authors develop a fine-tuning strategy for the VQ-VAE's decoder to create a noise-resistant one to reduce the gap between the data in training and real usages. A user study demonstrates the effectiveness of the proposed approach.

**Strengths:**

- originality-wise: the choice of converting all representations to a point cloud and using the point cloud to generate corresponding mesh is interesting;
- quality-wise: qualitative results are of high quality;
- clarify-wise: the paper is well-written and easy to follow;
- significance: to produce a mesh similar to what artists create is of great importance to bridge the current 3D generation research into real-world usage.

**Weaknesses:**

## Concerns about the user study

Since the goal of the project is to produce artist-created meshes, it is important to make sure that the model's output is aligned with real artists' preferences. However, there is no information about whether the 41 users who participated in the study (L517) are qualified especially Tab. 1 is the main results reported in the paper. For "qualified" users, I mean people who are artists with enough experience in the field of 3D creation or editing, etc.

Can authors clarify? If the users are not qualified, I would not be convinced by the result.

## Concerns about comparison with mesh generation pipeline

In L508, the authors state:
> MeshAnything is a shape-conditioned mesh generation method, we sampled shapes randomly from the evaluation set of Objaverse as inputs for MeshAnything, while for the baseline methods, we performed random sampling directly.

Further, in L862, the authors state:
> We quantitatively evaluate mesh quality by uniformly sampling 100K points from the faces of both the ground truth meshes and the predicted meshes, and then computing a set of metrics to assess various aspects of the reconstruction.

If I understand correctly:
- the procedure for evaluating MeshAnything is: 1) sample a mesh and then sample points from the evaluation set; 2) run MeshAnything on the sampled point cloud to obtain a predicted mesh; and 3) sample points from the predicted mesh.
- the procedure for the baselines are 1) directly run the generation; and 2) sample points from the generated mesh.

Such a setup is quite unfair in my opinion. Essentially, MeshAnything generates meshes aligned with the evaluation dataset. I think it will surely perform well in terms of those metrics for evaluating generation qualities in Tab. 2. Especially on Objaverse, which has tons of various categorical objects, the unfairness will be amplified.

I also agree that it is not easy to compare with baselines as MeshAnything needs to have a point cloud conditioning. ShapeNet may be more suitable due to the constrained categories.

## Questions about generation procedure

Can the authors clarify how the generation is conducted? As a generative framework, should the model output various plausible meshes based on the input point cloud? However, in the paper, there is always one mesh corresponding to the input. Does this mean that the authors always use **max** logits during the sampling, i.e., greedy search? Can authors provide more generation variants, e.g., with beam search? I would like to know how diverse the mesh generation could be.

Further, a question related to the evaluation on Tab. 2 is whether the authors only use **max** sampling to evaluate.

## Insufficient qualitative results

There are actually no qualitative comparisons to baselines, e.g., MeshGPT and PolyGen in the paper. Please provide.

## Citation format

Please refer to the template instructions and use the correct citation commands. Currently, all citations are with `\citet` instead of `\citep`. They look so weird as those citations appearing without parenthesis break the sentences.

**Questions:**

See "weakness".

---

> ### Author Response · Authors · 2024-11-17
> **Response to reviewer qFPE**
>
> We thank reviewer qFPE for the insightful comments and constructive feedback. Please see our response to the feedback below.
>
> ### 1. Concerns about the User Study (Responding to R3W1)
>
> The majority of participants in our user study were researchers from the 3D field, with a smaller portion being industry practitioners. We appreciate the reviewer for raising this point and have added this clarification to our updated paper (Page 10, Line 503).
>
> ### 2. Concerns about Comparison with Mesh Generation Pipeline (Responding to R3W2)
>
> Many thanks for your suggestion! Your understanding of our experimental procedure is completely accurate. We have also been exploring ways to make the comparison experiment fairer, and your suggestion has greatly helped to resolve this issue. The table below presents results from an additional experiment on ShapeNet with a fixed category, as you suggested. In this experiment, both MeshAnything and MeshGPT were trained and tested only on the "chair" category in ShapeNet.
>
> | Method         | **COV↑** | **MMD↓** | **1-NNA↓** | **FID↓** | **KID↓** |
> |----------------|----------|----------|------------|----------|----------|
> | MeshGPT        | 49.2     | 2.98     | 61.0       | 16.4     | 2.04     |
> | MeshAnything   | **62.1** | **1.92** | **49.8**   | **11.2** | **1.21** |
>
> It is worth noting that MeshGPT does not currently provide open-source code or data, so our experiments used a reproduced version of MeshGPT. To ensure a fair comparison, we reproduced MeshGPT with the exact same architecture and data as MeshAnything, differing only in the use of unconditional generation versus shape-conditioning.
>
> We observed that the gap between MeshGPT and MeshAnything narrowed under this setting, though a noticeable difference remains. This result supports our argument in Section 3 that shape-conditioned mesh generation is considerably less challenging to learn than unconditional mesh generation.
>
> The above experiment have been added to our new version (Page 19, Line 1015).
>
> ### 3. Questions about the Generation Procedure (Responding to R3W3)
>
> All of our qualitative and quantitative experiments employed multinomial sampling, equivalent to a beam search with a beam size of 1, using top_k and top_p values set to 50 and 0.95, respectively. Please see the Figure 8 of our newly updated (Page 20, line 1026), we provide generation variants obtained using this sampling approach.
>
> It is worth noting that generation using max logits tends to yield slightly higher success rates, making it preferable in certain cases.
>
> ### 4. Insufficient Qualitative Results (Responding to R3W4)
>
> We thank the reviewer for this suggestion and have added qualitative comparisons with baselines, such as MeshGPT and PolyGen, in our new version (Page 20, Line 1062).
>
> ### 5. Citation Format (Responding to R3W5)
>
> We apologize for the formatting oversight and thank the reviewer for bringing it to our attention. We have corrected the citation format in the main text to ensure consistency with the template instructions.

---

> ### Author Response · Authors · 2024-11-22
>
> We hope our reply could address your questions. As the discussion phase is nearing its end, we would be grateful to hear your feedback and wondered if you might still have any concerns we could address. It would be appreciated if you could raise your score on our paper. We thank you again for your effort in reviewing our paper.
>
> Best regards,
>
> MeshAnything Authors

---

> > ### Comment · Reviewer_qFPE · 2024-11-25
> >
> > I thank the authors' time and effort in addressing my concerns.
> >
> > After carefully reading other reviewers' comments, I lean towards maintaining my current score. On the positive side, I appreciate the work's effectiveness in generating "Artist-Created" meshes. However, I also understand the questions raised by other reviewers about scalability.

---

> ### Comment · Reviewer_qFPE · 2024-11-25
> **Change my score due to violation of anonymity policy**
>
> After another round of carefully reading of other reviews, I think the author's response to `Reviewer tUZi` violated the anonymity policy. Therefore, I cannot maintain my positive score.
>
> Specifically, the following comments under `3. Scalability of the Method (Responding to R2W3)` violate the anonymity policy as they effectively reveal the identities of the authors:
>
> > In fact, recent works [1,2,3] following our method have successfully scaled up.
> > 1. Edgerunner: Auto-regressive Auto-encoder for Artistic Mesh Generation. Tang et al., 2024. https://arxiv.org/abs/2409.18114
> > 2. Meshtron: High-Fidelity, Artist-Like 3D Mesh Generation at Scale. Anonymous, 2024. https://openreview.net/forum?id=mhzDv7UAMu https://meshtron.github.io/index.html
> > 3. Scaling Mesh Generation via Compressive Tokenization. Weng et al., 2024. https://arxiv.org/abs/2411.07025

---

> > ### Author Response · Authors · 2024-11-26
> >
> > We sincerely thank Reviewer qFPE for their careful review.
> >
> > However, we are unclear about why Reviewer qFPE believes we have violated the anonymity policy. Edgerunner and Scaling Mesh Generation via Compressive Tokenization are preprints on arXiv, while Meshtron is an anonymous submission to ICLR 2025. To facilitate easy access, we have provided hyperlinks.
> >
> > Could Reviewer qFPE kindly elaborate on how we might have violated the policy?

---

> ### Comment · Reviewer_qFPE · 2024-11-26
>
> Thanks for asking. Those papers do not cite your MeshAnything work in an anonymous way. Since you mentioned that they are your follow-up works, you essentially reveal your identities. Namely, anyone who searches the word "MeshAnything" in those papers will know your identities. I hope this makes things clear.

---

> ### Author Response · Authors · 2024-11-26
> **Response to Reviewer qFPE**
>
> We thank Reviewer qFPE for the valuable feedback.
>
> 1. The purpose of citing these papers is solely to demonstrate the scalability of our method. Apart from citing these papers as evidence, we have no better way to substantiate this point. In the rebuttal, we simply provided links to the arXiv papers and did not direct reviewers to read any author-identifying information. ICLR explicitly allows the citation of arXiv papers, which may naturally contain information about the authors of the submission. After thorough review, we found no ICLR guideline prohibiting the citation of papers that reference the submitted work. Therefore, we believe we have not violated the anonymity policy.
>
>
> 2. The ICLR 2019 Reviewer Guidelines ([link](https://iclr.cc/Conferences/2019/Reviewer_Guidelines#:~:text=Anonymity.,violation%2C%20contact%20your%20AC%20immediately.)) state (We failed to find more recent detailed guidelines regarding anonymity and hope the AC can provide clarification.):
>    > Anonymity. ICLR is double-blind, which means that authors are not aware of reviewer identities and reviewers are not aware of author identities. If you believe a paper contains an anonymity violation, contact your AC immediately.
>    > **Anonymity violations are not considered as part of reviewing criteria, they are requirements for submission. Unless your AC decides that the paper does indeed violate anonymity, proceed to review it as normal.**
>
>       Based on this guideline, we kindly believe that whether we violated the anonymity policy should be determined by the AC, and a possible violation should not serve as a reason for Reviewer qFPE to lower our score. We have contacted the AC regarding this matter and respectfully request that Reviewer qFPE consider restoring the previous score of 6.

---

### Official Review · Reviewer_tUZi · 2024-11-02

**Soundness:** 3
**Presentation:** 4
**Contribution:** 3
**Rating:** 5
**Confidence:** 5

**Summary:**

This paper introduces Meshanything for shape-conditioned AM generation. Meshanything first trains a VQ-VAE to obtain a set of mesh tokens and employs a noise-resistant decoder to enhance mesh generation quality. Subsequently, a shape-conditioned autoregressive transformer is trained to generate artist-created meshes. Experiments show that Methanything outperforms previous mesh generation methods with fewer faces.

**Strengths:**

- Mesh generation toward artist creation has been seldom explored in previous research. This work could serve as an improved remeshing tool, facilitating a more rational topology for the generated 3D assets.

- The visualization is good. The final mesh reconstructions are nice looking, and the method outperforms previous methods in the experiments.

- This work constructs coarse meshes for sampling point clouds as inputs, thereby narrowing the gap between training and inference. This approach is reasonable.

**Weaknesses:**

- This paper mainly focuses on the task of point cloud -> mesh, which is usually referred to as mesh reconstruction, rather than mesh generation. Though a transformer-based autoregressive architecture is applied, I still don't think it is appropriate to name the paper a mesh generation paper.

- It is unfair to use meshes generated by Rodin as the shape conditioning for comparison with MeshGPT. From my understanding, the Rodin engine largely (or purely) provides the generation ability while MeshAnything only acts as a remesher component.

- Another concern is that the number of AM mesh faces generated by MeshAnything is limited to a maximum of 800. Using such a limited number of faces makes it insufficient for representing meshes with more complex structures.

- This work appears to have stringent data requirements, as the authors were only able to filter 56k high-quality artist-created meshes from the Objaverse and ShapeNet datasets for training. So it may be challenging to scale this method up to more data and modeling parameters (but it might be sufficient to train a point cloud reconstructor/mesh remesher).

**Questions:**

My suggestion is to change the description of mesh generation to mesh reconstruction/remeshing.

I regard the proposed method as a novel learning-based remesher (AR applied), but I don't think the paper could be classified into a generation paper.

---

> ### Author Response · Authors · 2024-11-17
> **Response to reviewer tUZi**
>
> We thank reviewer tUZi for the insightful comments and constructive feedback. Please see our response to the feedback below.
>
> ### 1. Whether This Work is a Generation Model (Responding to R2W1)
>
> We appreciate the reviewer’s suggestion regarding the classification of our method as a generation model. While we acknowledge that our method shares some similarities with reconstruction approaches, we maintain that it fundamentally belongs in the generative model category.
>
> The generative nature of our approach is primarily evident in its flexibility with topology. Please see the Figure 8 of our newly updated (Page 20, line 1026), by varying the random seed under the same point cloud condition, our model generates multiple meshes with diverse topologies—something reconstruction methods cannot achieve.
>
> ### 2. It is unfair to use meshes generated by Rodin as the shape conditioning for comparison with MeshGPT. (Responding to R2W2)
>
> MeshGPT does not currently provide open-source code or data. To ensure a fair comparison, we reproduced MeshGPT with the exact same architecture and data as MeshAnything, differing only in the use of unconditional generation versus shape-conditioning. The purpose of this comparison is to contrast these two settings to support our claim in Section 3: shape-conditioned mesh generation significantly reduces the complexity of mesh generation by avoiding the need to learn a complex 3D shape distribution.
>
> Our method’s substantial advantage over MeshGPT in this comparison comes from its avoidance of learning a complex 3D shape distribution, instead relying on shape information provided by methods such as Rodin. This distinction is a core contribution of our work and the motivation behind our comparison with MeshGPT.
> .
> Besides, we agree that MeshAnything can be viewed as a remeshing model, with its generative ability reflected in topology rather than shape.
>
> ### 3. Scalability of the Method (Responding to R2W3)
>
> Our model generates meshes with a maximum of 800 faces due to computational constraints. However, as our method utilizes autoregressive transformer, it holds potential for scaling similar to large language models. Numerous techniques from the LLM domain help mitigate memory issues, avoiding the quadratic scaling of memory with mesh face numbers. In fact, recent works [1,2,3] following our method have successfully scaled up; for instance, Meshtron[2] now supports up to 64K faces.
>
> ### 4. Stringent Data Requirements (Responding to R2W4)
>
> Our filtering process resulted in a dataset of 56K samples primarily due to our face-count limitation. In the Objaverse and ShapeNet datasets, approximately 64K triangle meshes have fewer than 800 faces, and we filtered out only about 10% of these to remove redundant or overly simplistic data. With recent advances, such as the extension of the maximum face count to 64K in Meshtron[2], over 80% of the data (about 70K meshes) in Objaverse and ShapeNet would become usable for our approach.
>
> ### References
> 1. Edgerunner: Auto-regressive Auto-encoder for Artistic Mesh Generation. Tang et al., 2024. [https://arxiv.org/abs/2409.18114](https://arxiv.org/abs/2409.18114)
> 2. Meshtron: High-Fidelity, Artist-Like 3D Mesh Generation at Scale. Anonymous, 2024. [https://openreview.net/forum?id=mhzDv7UAMu](https://openreview.net/forum?id=mhzDv7UAMu) [https://meshtron.github.io/index.html](https://meshtron.github.io/index.html)
> 3. Scaling Mesh Generation via Compressive Tokenization. Weng et al., 2024. [https://arxiv.org/abs/2411.07025](https://arxiv.org/abs/2411.07025)

---

> > ### Comment · Reviewer_tUZi · 2024-11-26
> >
> > I thank the authors for their responses. I have no further concerns about the scalability part, however, I am still not convinced by the explanation of reconstruction v.s. generation.
> >
> > - Varying topologies of the same shape can also be achieved by changing methods/parameters in traditional reconstruction/re-meshing approaches.
> > - The authors claim the advantage of the proposed method comes from "its avoidance of learning a complex 3D shape distribution". However, this contradicts my understanding of 3D generation directly. From my perspective, the core of 3D generation is to learn the 3D shape (and sometimes appearance) distribution, and this is why I think the Rodin engine largely (or purely) provides the generation ability. Thus, I cannot buy the generation story.

---

> > > ### Author Response · Authors · 2024-11-26
> > > **Response to Reviewer tUZi**
> > >
> > > We thank reviewer tUZi for the insightful feedback.
> > >
> > > 1. We agree that reconstruction-based methods can generate different topological results by adjusting parameters. However, compared to the meaningful topology of Artist-Created Meshes, the topology from reconstruction-based methods can be regarded as noise, lacking any information. While reconstruction-based methods and remeshing techniques can modify topology, these changes are typically transitions from one form of noise to another.
> > >
> > > 2. We agree that learning 3D shape distributions is a core aspect of 3D generation. However, we believe that topology generation (i.e., shape-conditioned mesh generation introduced in Section 3 of the main paper) is also crucial for advancing 3D generation. Previous works have largely overlooked topology generation, despite topology being a critical property of meshes—one of the most widely used 3D representations. Topology significantly impacts both the quality and efficiency of meshes, which are essential for the application of 3D generation in the 3D industry. Methods like Rodin, which focus on generating 3D shapes, are not directly comparable to MeshAnything. Instead, their outputs should be viewed as the input to MeshAnything.

---

> ### Author Response · Authors · 2024-11-22
>
> We hope our reply could address your questions. As the discussion phase is nearing its end, we would be grateful to hear your feedback and wondered if you might still have any concerns we could address. It would be appreciated if you could raise your score on our paper. We thank you again for your effort in reviewing our paper.
>
> Best regards,
>
> MeshAnything Authors

---

### Official Review · Reviewer_7tHF · 2024-11-03

**Soundness:** 2
**Presentation:** 3
**Contribution:** 2
**Rating:** 5
**Confidence:** 3

**Summary:**

This work introduces a method for conditional "artist-created" mesh generation.
The key idea is employing auto-regressive transformer on top of a VQ-VAE pre-trained latent space, with additional conditioning for VQ-VAE and noise tricks to improve generalization. Quantitative results on toy datasets suggest that the method outperforming recent baselines.

**Strengths:**

+ The paper is well-written and is easy to follow.
+ The overall architecture of the method makes sense, and the problem of conditional mesh generation is important.
+ There are several interesting techniques that could be useful for practitioners: e.g. conditioning VQ-VAE on the shape to improve reconstruction, and the gumbel noise trick to improve VQ-VAE's decoder robustness.
+ Quantitative results (Table 2) suggest that the method outperforms competitors.

**Weaknesses:**

- The motivation for focus on artist-generated meshes is not very clear - would exactly the same method not work on a collection of reconstruction-based meshes? My assumption would be that the reason is that the method cannot scale to any realistic number of vertices.
- The method is a combination of existing architectures (VQ-VAE based on BERT + OPT transformers), whereas the encoding scheme is the same as in polygen / meshgpt.
- The scalability of the method is very questionable: both in terms of training and inference. If my undertstanding is correct, memory scales quadratically with the number of mesh faces, and authors explicitly mention that they filter out meshes with less than 800 faces. There is discussion of this in the appendix, but it is worth providing more information (memory and/or runtime numbers) which would be critical to get a complete impression of how usable the method actually is.
- Comparison to diffusion-based methods (e.g. point-e) would be interesting (converting point clouds to meshes should be supported by publicly available code).

**Questions:**

- Can you provide details on the scalability of the method - i.e. what is the max number of faces/vertices the model can handle?
- Is there anything specific about architecture why artist-generatedness of the meshes is critical?
- How does this method stack against diffusion-based methods for mesh/point-cloud based generation (e.g. point-e)?

---

> ### Author Response · Authors · 2024-11-17
> **Response to reviewer 7tHF**
>
> We thank reviewer 7tHF for the insightful comments and constructive feedback. Please see our response to the feedback below.
>
> ### 1. Scalability of the Method (Responding to R1W1, R1W3, and R1Q1)
>
> Our model generates meshes with a maximum of 800 faces due to computational constraints. However, as our method utilizes autoregressive transformer, it holds potential for scaling similar to large language models. Numerous techniques from the LLM domain help mitigate memory issues, avoiding the quadratic scaling of memory with mesh face numbers. In fact, recent works [1,2,3] following our method have successfully scaled up; for instance, Meshtron[2] now supports up to 64K faces.
>
> ### 2. Comparison to Point Cloud Generation Methods (Responding to R1W4 and R1Q3)
>
> We appreciate reviewer 7tHF’s suggestion and have added a comparison experiment. Since Point-E primarily generates point clouds from text, while our method generates Artist-Created Meshes aligned with the input point cloud, a direct comparison is challenging. When applying methods like Point-E in industry, generated point clouds often need to be converted to meshes, making these point cloud-to-mesh conversion methods more comparable to our approach.
>
> Therefore, we selected Poisson surface reconstruction, one of the most widely used point cloud-to-mesh methods, to conduct the following experiment: we sampled a dense point cloud of 100K points on each ground truth mesh surface, reconstructed it using Poisson surface reconstruction, and then optimized the face count through Blender Remesh. The results are presented below.
>
> | Method                    | **CD↓**   | **ECD↓**  | **NC↑**   | **#V↓**   | **#F↓**   | **VR↓**   | **FR↓**   |
> |---------------------------|-----------|-----------|-----------|-----------|-----------|-----------|-----------|
> | MeshAnything              | 2.256     | **6.245** | 0.902     | **0.172** | **0.318** | **0.888** | **0.871** |
> | Poisson + Remesh (0.010)  | **2.105** | 7.655     | **0.936** | 39.29     | 42.39     | 227.1     | 133.6     |
> | Poisson + Remesh (0.050)  | 4.355     | 8.435     | 0.831     | 2.383     | 1.615     | 14.28     | 5.166     |
>
> From the above results, point cloud to mesh methods yield outcomes similar to those produced by Marching Cubes, making it challenging for them to achieve both efficiency and quality in mesh generation. We have included this experiment in the main text.
>
> ### 3. Importance of Artist-Created Meshes (Responding to R1Q2)
>
> Compared to reconstruction-based meshes, Artist-Created Meshes (AMs) offer significant advantages in terms of efficiency and aesthetics.
>
> First, converted meshes often contain orders of magnitude more faces than AMs, leading to inefficiencies in storage, rendering, and simulation. Additionally, their complex and chaotic topologies make post-processing and downstream tasks in the 3D pipeline more challenging, requiring extensive optimization by human artists. Finally, reconstruction-based methods struggle to capture sharp edges and flat surfaces, resulting in oversmoothing and bumpy artifacts, as shown in Fig. 2 of our main paper. These factors make reconstruction-based meshes less suitable for applications, such as 3D games, where efficient rendering and high visual quality are essential.
>
> We kindly direct reviewer 7tHF to line 88 on page 2 of the main text for further details.
>
> ---
>
> ### References
> 1. Edgerunner: Auto-regressive Auto-encoder for Artistic Mesh Generation. Tang et al., 2024. [https://arxiv.org/abs/2409.18114](https://arxiv.org/abs/2409.18114)
> 2. Meshtron: High-Fidelity, Artist-Like 3D Mesh Generation at Scale. Anonymous, 2024. [https://openreview.net/forum?id=mhzDv7UAMu](https://openreview.net/forum?id=mhzDv7UAMu) [https://meshtron.github.io/index.html](https://meshtron.github.io/index.html)
> 3. Scaling Mesh Generation via Compressive Tokenization. Weng et al., 2024. [https://arxiv.org/abs/2411.07025](https://arxiv.org/abs/2411.07025)

---

> ### Author Response · Authors · 2024-11-22
>
> We hope our reply could address your questions. As the discussion phase is nearing its end, we would be grateful to hear your feedback and wondered if you might still have any concerns we could address. It would be appreciated if you could raise your score on our paper. We thank you again for your effort in reviewing our paper.
>
> Best regards,
>
> MeshAnything Authors

---

> ### Comment · Reviewer_7tHF · 2024-11-22
>
> I thank authors for their clear response!
>
> 1. scalability of the method
> It does sound like 800 faces is probably insufficient for a lot of realistic applications. I think this is particularly making paper's claims about efficiency weaker.  I am not sure if we can judge the scalability of this work based on scalability of derived works.
>
> 2. comparison to point cloud
> agreed, direct comparison is challenging.
>
> 3. importance of artist-created meshes
> Some of the arguments are valid - in particular about chaotic topologies. Some are very arguable - e.g. reconstruction-based method being unable to capture sharp details - this is really setup dependent. I guess the question is more about method being in any way specific to artist meshes - as paper writing suggests so, but in my humble opinion those are completely independent.
>
> In summary, I still think the scalability of the method is a major issue. Additionally, the main message of the paper remains unclear to me - in particular limiting application of a seemingly more generic method to artist meshes only. Thus I am inclined to keep my original rating.

---

> > ### Author Response · Authors · 2024-11-23
> >
> > We thank Reviewer 7tHF for the insightful feedback.
> >
> > ## Scalability of the Method
> >
> > We agree that 800 faces are insufficient for realistic applications. However, we argue that this limitation primarily stems from computational resource constraints: MeshAnything was trained for only 4 days on 8 A100 GPUs. Recently, Meshtron introduced efficient training and inference techniques from the LLM domain into the shape-conditioned mesh generation paradigm we proposed in Section 3, along with greater computational resources and data, successfully scaling up to 64000 faces. Specifically, Meshtron achieved this by using truncated training to reduce memory requirements during training and applying sliding window attention during inference to lower computational and memory demands.
> >
> > In terms of fundamental settings, Meshtron remains consistent with the shape-conditioned mesh generation framework introduced in this work, with differences mainly in the engineering implementations for large-scale training. Therefore, we argue that Meshtron’s scaling results can serve as a reference for the scalability potential of our approach.
> >
> > ## Importance of Artist-Created Meshes
> >
> > Artist-Created Meshes excel in both efficiency and high-quality topology. While reconstruction-based methods can capture sharp details, they do so at the expense of extremely dense face counts. For example, a simple tetrahedron can be represented with just 4 faces in an artist-created mesh. In contrast, reconstruction-based methods can continually increase resolution to approximate the tetrahedron, but they can never achieve a perfectly precise representation.
> >
> > The goal of our work is not to entirely replace reconstruction-based methods, which still have advantages in certain scenarios (e.g., higher success rates and shorter inference times). Instead, we aim to propose a new direction for generating Artist-Created Meshes from various conditions, reducing the need for extensive human labor in fields such as gaming, film production, and 3D simulation.

---

### Comment · Area_Chair_cVdH · 2024-11-25
**Last day for interactive discussions!**

Dear authors and reviewers,

The interactive discussion phase will end in one day (November 26). Please read the authors' responses and the reviewers' feedback carefully and exchange your thoughts at your earliest convenience. This would be your last chance to be able to clarify any potential confusion.

Thank you,
ICLR 2025 AC

---

> ### Author Response · Authors · 2024-11-26
> **Request for Review of Anonymity Violation**
>
> Dear AC,
>
> Reviewer qFPE raised a concern about a potential anonymity violation in our rebuttal and lowered the score from 6 to 3 based on this. We respectfully request the AC to make a determination on this matter and suggest that ICLR consider providing more detailed guidelines regarding anonymity policies, specifically clarifying whether it is permissible to cite papers that reference the paper under review.
>
> Thank you,
>
> MeshAnything Authors

---

### Meta-Review · Area_Chair_cVdH · 2024-12-20

**Metareview:**

The submission received somewhat mixed reviews. The reviewers appreciate the work addressing artist-like shape generation for practical artistic applications, and strong results were presented. The major concerns from 7tHF and tUZi were on doubts on scalability and missing/unfair comparisons. The AC carefully read through the paper, the reviewers' comments, the authors' rebuttal and the discussions. While reservations for scalability remains, the contributions of the paper includes a new method that address a practical problem with strong preliminary results. The AC believes the merits outweighs the limitations in general; as such, the AC recommends acceptance.

**Additional Comments On Reviewer Discussion:**

The reviewers raised questions mostly regarding scalability (7tHF, tUZi), missing/unfair comparisons (7tHF, tUZi, qFPE, 3Tqd), and task framing of reconstruction vs. generation (tUZi). Concerns around scalability and comparisons have been partially addressed. On tUZi's question on reconstruction vs. generation, the AC agrees with the authors that the work can be considered a 3D generative model. The AC disregards any discussion on citing follow-up works during this process.

---

### Decision · Program_Chairs · 2025-01-22

Accept (Poster)